# Self-Aware Cybersecurity Architecture for Autonomous Vehicles: Security through System-Level Accountability

**DOI:** 10.3390/s23218817

**Published:** 2023-10-30

**Authors:** Akwasi Adu-Kyere, Ethiopia Nigussie, Jouni Isoaho

**Affiliations:** Department of Computing, University of Turku, Vesilinnatie 5, 20500 Turku, Finland; ethiopia.nigussie@utu.fi (E.N.); jouni.isoaho@utu.fi (J.I.)

**Keywords:** security, cybersecurity, autonomous vehicles, connected vehicles, self-aware architecture, intelligent vehicles, security architecture, security accountability

## Abstract

The inherent dynamism of recent technological advancements in intelligent vehicles has seen multitudes of noteworthy security concerns regarding interactions and data. As future mobility embraces the concept of vehicles-to-everything, it exacerbates security complexities and challenges concerning dynamism, adaptiveness, and self-awareness. It calls for a transition from security measures relying on static approaches and implementations. Therefore, to address this transition, this work proposes a hierarchical self-aware security architecture that effectively establishes accountability at the system level and further illustrates why such a proposed security architecture is relevant to intelligent vehicles. The article provides (1) a comprehensive understanding of the self-aware security concept, with emphasis on its hierarchical security architecture that enables system-level accountability, and (2) a deep dive into each layer supported by algorithms and a security-specific in-vehicle black box with external virtual security operation center (VSOC) interactions. In contrast to the present in-vehicle security measures, this architecture introduces characteristics and properties that enact self-awareness through system-level accountability. It implements hierarchical layers that enable real-time monitoring, analysis, decision-making, and in-vehicle and remote site integration regarding security-related decisions and activities.

## 1. Introduction

Today, vehicles as integral components of cyber-physical systems (CPS), are anticipated to effectively address the pressing automobile challenges. As future mobility embraces the concept of vehicles-to-everything, it exacerbates security complexities and challenges [1,2] concerning dynamism, adaptiveness, and self-awareness. The inherent fluidity of these technological advancements in intelligent mobility has seen multitudes of noteworthy security improvements. However, there is still a call for a transition from security measures dependent on static approaches and implementations. For example, in such a transition, the vehicles utilized imply that open software protocols and connections for vehicle and electric in-vehicle infrastructure are critical to their security posture. Couple these with a collection of evolving sensor platform vehicles and they would become fully functional complex computers on wheels [3]. Such capabilities will facilitate in-vehicle computing, fleet management [4], synchronization, telemetry, and the bidirectional sharing of information [5] in urban mobility. They will also be capable of handling network traffic transactions through embedded systems [6]. Characteristics and properties such as these make connected, intelligent, and autonomous vehicles social “things” capable of generating enormous amounts of data and inadvertently acting as virtual data sources on wheels and intelligent mobile objects [4].

### 1.1. Background

The traits and characteristics of security are fundamental to the atomic reality of connected, intelligent, and automated vehicles because of its crucial role in shaping [7] and influencing associating technologies [8]. The issue of security is a persistent and influential factor that gradually influences the landscape of automotive technologies, particularly in the context of the interplay between and from conventional driving and autonomous driving system security. With the widespread agreement and consensus of automobiles adhering to cybersecurity regulations as security controllers, cybersecurity solutions of varying complexity and scope are widely accessible across many professions and domains [9]. Specific security requirements covering areas, such as application security, intrusion detection prevention systems (IDS/IPS), secure over-the-air (OTA) transactions, trusted execution environments (TEE), in-built hardware security, and root-of-trust [10], are considered part of the primary security measures in today’s modern automobiles.

### 1.2. Research Objectives, Motivation, and Contributions

The diverse security requirements of connected, intelligent, and autonomous vehicles call for a systematic and robust approach that integrates and facilitates seamless interaction among solutions [11]. In contrast to today’s widespread in-vehicle security measures, a hierarchical self-aware cybersecurity architecture consisting of real-time security monitoring, analysis, decision-making, and visualization layers is due for this transition. The security characteristics and properties must also enforce security through system-level accountability. Therefore, this work proposes a self-aware security with architectural sub-modularization components. The proposed cybersecurity architecture addresses the inadequacy of security dynamism, adaptiveness, and self-awareness in the vehicular ecosystem. It provides (1) a comprehensive understanding of the self-aware security concept, with emphasis on its hierarchical security architecture that enables system-level accountability, and (2) a deep-dive into each layer supported by algorithms and a security-specific in-vehicle black box with external VSOC interactions. In contrast to the present in-vehicle security measures, this architecture introduces characteristics and properties that enact self-awareness through system-level accountability. It implements hierarchical layers that enable real-time monitoring, analysis, decision-making, and in-vehicle and remote site integration regarding security-related activities. Hence, the contributions of this work are listed as follows:A systematic exploration of security dynamism, self-awareness, and adaptiveness is provided in the vehicular ecosystem. It examines in-depth the need for appropriate security measures that match the dynamic nature of the vehicular security landscape.A hierarchical self-aware security architecture is proposed that effectively establishes accountability at the system level and further illustrates why such a proposed security architecture is relevant to intelligent vehicles.The sub-components of the self-aware architecture, their algorithms, and the integration of a security-specific black box are presented.

### 1.3. Scope of This Research Work

The scope of this research manuscript is to share a study that proposes a security architecture. The underlying target seeks to address the inadequacies of dynamic, self-aware, and adaptive security measures applicable to connected, intelligent, and autonomous vehicles. Subsequently, the following set of security assumptions is fundamental to the proposed self-aware security architecture. The first assumption asserts that the connection between interconnected nodes is encrypted. Secondly, it assumes that decryption is applicable to all indicated data flow from one point to another and is decrypted when necessary. Lastly, we assume that all illustration model arguments are from a security context and apply to connected, intelligent, and autonomous vehicles. The remaining sections of this work have been grouped as follows. Section 2 provides a literature review and an overview of the current solutions, serving as the contextual background. Section 3 of the paper presents an exposition and elucidation of the suggested self-aware security architecture, as well as an explication of the structure of the security architecture. Section 4, Section 5, Section 6, Section 7 and Section 8 provide a comprehensive deep dive into the design aspects of each individual component. Section 9 presents a discussion leading to the conclusion in Section 10.

## 2. Literature Review

When the future of transportation began envisioning vehicles (connected, intelligent, and autonomous) that can communicate with intelligent gadgets, the Internet of Things (IoTs), and other external infrastructures, security became a significant issue. Recent projections predict 115 million connected vehicles (CVs) by 2025, while the frequency of attacks on modern cars has increased by 225% in the past three years [12]. These vehicular networks will rely on complex interactions between vehicles, intelligent gadgets, the Internet of Things (IoTs), people, and external infrastructure (vehicle-to-everything (V2X)). Research into these security concerns has seen a focus on influence and acceptance factors, cybersecurity challenges and evaluations, vulnerability, sophistication, cybersecurity risk management, threat modeling, and susceptibility. In addition to these, a substantial effort is also seen in addressing many of these security challenges in modern vehicles with the use of security measures such as trusted execution environments (TEEs) [13,14], white-listing, tamper-proofing, access control [15], and others [16,17,18,19,20], with details illustrated in Table 1.

The current automobile security infrastructure state has seen further research in several security sectors with increasing demand for vehicular services [30], vehicular network deployments [31], and in-vehicle electric infrastructure [32]. These research aspects generate comprehensive insights into the fast-approaching ubiquitousness of connected, intelligent, and autonomous ecosystems. However, despite the substantial anticipation of addressing these security challenges in modern vehicles by various vendors, the intensity, diversity, and impact percentage of attacks have increased in the past few years [12]. Nevertheless, several lines of evidence from these examples in Table 1 show that various security implementations, including intrusion detection systems, two-factor authentication, access control mechanisms, TEE storage and encryption, tamper-proofing, secure boot, white-listing, and firewalls, have already been deployed by several vendors. Examples include Nissan, BMW, Audi, Toyota, Honda, Mercedes-Benz, General Motors, Tesla, Volkswagen, Porsche, Daimler Trucks, Jaguar Land Rover (JLR), Ford, Mitsubishi, Volvo Trucks, Subaru, and Hyundai. However, few security solutions compensate for the vehicle’s lifespan and continuity.

Research such as [33,34,35,36,37] has paid great attention to influence and acceptance factors. In contrast, [8,38,39,40,41,42,43,44,45,46] have focused on cybersecurity challenges and evaluations. Drawing parallelism from cybersecurity, risk assessment, security management techniques, security impacts and effects, and threat analysis has been beneficial for quantifying security. Besides proprietary vendor cybersecurity solutions deployed in modern vehicles in Table 1, further substantial literature and exploratory investment from theories, concepts, frameworks, and experimental proposals demonstrating vulnerabilities include [44,47,48] with sophistication and susceptibility across processes and sensors.

In [49], the authors presented a cybersecurity and safety management system (CSSMS) approach relying on the ISO 26262 and ISO/SAE 21434 for integrated safety and cybersecurity process management in vehicles. Alladi et al. [50] proposed a lightweight and secure authentication and attestation scheme for attesting dynamic transit vehicles in line with [51]. Khan et al. [52] proposed a real-time intrusion detection framework based on normal state-based and a deep learning-centered bidirectional long short-term memory (LSTM) architecture. These studies have recognized the importance of security, its outcomes, and specific aspects. However, security implementations, architectures, and schemes integrating specific solutions capturing the complex interaction within the context of connected, intelligent, and autonomous vehicles still need further research. Understanding the risk and technological feasibility is one part, while extrapolating or developing technical security solutions is another. The considerable amount of research on automotive cybersecurity is vast and diverse to the point that the security scheme depends on the implementor or vendor. Furthermore, the transcendency in the gravity of security for vehicles (connected, intelligent, and autonomous) in the cybersecurity beneficiary often relies on the ability to detect and determine security threats through monitoring scenarios.

## 3. Proposed Self-Aware Security Architecture

The proposed hierarchical self-aware security architecture extends the state-of-the-art illustrated in Figure 1 by providing dynamic monitoring, analysis, decision-making, visualization, and the capability to integrate various security solutions with security-specific black box. In its generic form in Figure 2, the architecture consists of (1) distributed monitoring agents, monitored components (e.g., services, processes, and network transactions), and data feeder at the monitoring layer, (2) process controllers at the analysis layer and decision controllers at the decision layer. Any services, processes, and communication are monitored by the agents and analyzed by the process controllers. A set of decision controllers act on the information from the process controllers. The decisions are archived in the black box, while the analysis, report, and visualization layers are capable of both in-vehicle and external virtual security operation center (VSOC) HMIs.

Connected, intelligent, and autonomous vehicles will generate real-time in-vehicle and external security system incidents. As complex relational systems, for example, the processes and activities with instantaneous in-vehicle or offloading compute probabilities must be distinguishable in their advocative realm for action augmentation. Figure 3 systematically illustrates this concept at the intake stage. Several layers of system-level accountabilities start with incident generation from the components and nodes related to security. Therefore, system-level accountability of relevant in-vehicle and external processes and activities’ security states must be known whenever needed in attaining self-awareness in a defensive security posture.

Hence, the reflective vehicular security state and its intricacies’ distinguishability start in the analysis module. The process controllers communicate with decision controllers about the security analysis results on current incidents, allowing the maintenance of the operational environment security status and awareness with the help of the decision controller. The most interesting aspect of the analysis module, which is essential to autonomy, is the capability of dynamic, iterative analysis, continuous monitoring of security events, and adaptability while rendering data from in-vehicle process nodes and devices. Other granular processing from the analysis module to decision modules is also derivable into pathways such as the virtual security operation center (VSOC) and in-vehicle critical incident archiving in the security-specific black box. With the help of agents, the black box intends to store these actions or sync with an external source.

### Architectural Component

The proposed architecture has four main components: monitoring, analysis, decision-making, and visualization. In this context, the term controller does not necessarily refer to the governed attached devices’ functions but emphasizes that the data are not mutable upon interactions. The process-based module controller has four module-based deployments: the analysis module, the fail-over mechanism module, the fall-back mechanism module, and the report/support module. The data intake feeders are the shippers of various metrics and incidents, such as measurements, status monitoring, and extracting data from controllers for scrutinization.

In a complex system, such as the central computing unit of an autonomous vehicle, processes and applications perform their routine operations by making changes. They interact with other processes and request to perform specific actions on behalf of other processes and applications translative to external and in-vehicle security interactions. The security system-call routines follow this perspective, deducing that the self-aware system-level security accountability instantiates these system-calls to their respective integrity levels. For example, critical instances of in-vehicle interactions with external instances report an answerable justification upon exploring and investigating these system-calls made by applications and processes of interest through systematic flow. This accountability is extendable to the system, user, and specific redefined space within the decision module-based controllers. For example, the three decision modules, namely, the system incident, critical, and resolvable module, evaluate security actions to permissions-related, processes-related, and predefined-related against pre-defined reporting incidents and rules (i.e., incidents, users, models, or within the model that triggered the rule). Command executions in a defined space enable the assessment of events from the analysis phase to the resolvable decision module. Furthermore, queries relating to specific instances of command executions and the support module mechanism visible in Figure 4 also relate to recording, keeping, and tracking summarizations.

## 4. Analysis

Figure 5 represents the analysis component from the analysis layer in Section 3. The scope covers pre-defined configurations, firewalls, and intrusion prevention and detection systems (IPS/IDS). Its functionality is to handle and analyze security-specific instances and incidents reported through the data feeders by the agents. As shown, analysis from the data intake feeders in the form of system and process activities, for example, internal services, runtime processes, applications, security audits, external event agents, infrastructural services, and external services, are sent to the analysis layer. The incident filtering and classification mapping is part of the analysis block. Based on the architectural implementation choice considering specific instances or applications, the analysis block can be implemented as a unit or as a supporting unit.

Part of this framework’s focal point is maintaining a dynamic security awareness through analytical parameters. An analysis is essential in most security implementations that depend or rely on dynamism via expressing action rather than maintaining a state and implying that static security measures and parameters need to evolve, especially with the advent of applicable artificial intelligence (AI) and quantum computers on the rise. In this case, achieving a dynamic security measure often involves incident filtering and classification mapping of internal and external service communications, application executions in a TEE, dynamic white-listing, and access controlling. However, resource constraints and implementation restrictions can dictate how comprehensive or compensating analysis is applied. In vehicles, for example, necessary procedures such as analysis are beneficial for multiple reasons, such as availability, consistency, and network connection restraints to reduce latency and response times. Furthermore, implementing such a concept in an electronic control unit (ECU) via platforms such as AUTOSAR’s classic platform [53], the Adaptive platform [54], or any other example will require associating implementation requirements.

Transactions and connectivity initiation traffic have a single ingress at the analysis layer. It includes runtime processes fulfilling the needs of security-specific devices or nodes such as firewalls, IPS/IDS, and auditing nodes depending on the task, purpose, or need. In state-of-the-art vehicular security, white-listing, access control, and firewall deployments exist from vendors. However, some of these implementations have precise tailoring to unique instances and situations lacking security dynamism in security based on the situation. Furthermore, with the lifespan of vehicles and continuity factors influencing security, connected, intelligent, and autonomous vehicles, firewall implementation, for example, needs a dynamic security mechanism that can be assistive via analysis at either the circuit level, application level, or packet filtering level [29]. The capability of making security decisions in dilemmas within the vehicular ecosystem and vehicle-to-everything internal and external communication and transaction is vital in such a security mechanism or solution.

## 5. Decision

In this section, a detailed account of the three decision module-based controllers in the decision layer, namely, the system incident, critical incident, and resolvable controller, which are already visible in both Figure 2 and Figure 4 and discussed in Section 3, is given. Each module-based control is comprehensively expressed with algorithms related to these decision modules and is also illustrated as Algorithms 1–3, respectively. The decision modules’ operation follows each other and carries the identifier decision phases, respectively, from (I) to (III) in the whole decision-making chain. Throughout the algorithms, the term classification explains the process whereby data from one phase to another are grouped based on the parameters and characteristic specifications of the decision module-based controller. The specification is dependent on the functionality of the decision module-based controller, which may vary across implementations.

### 5.1. System Incident Module-Based Controller

In decision phase (I), data from the analysis module controller enter the first decision module (system incident). Each entry initiates a security check that involves (1) checking the validity of that specific message, (2) checking the classification attributes, associated tags, and markings that are unique to individual components, such as sensors, secure gateways, infotainment, and others, and (3) checking if they are filtered to make sure the appropriate delivery message is to and from the required source-to-destination. Proceeding with these security measures follows the reporting measures that archive each decision taken to the BlackBox with the help of agents. The classified input data that pass the security checks proceed to the next decision phase visible in Algorithm 1, Lines (6–14). The outcome of this decision forwards data to the next phase or rejects/drops failed messages.
**Algorithm 1** Decision module for system incidence.
   **Input**: Raw data from analysis layer
   **Output**: System incident level—decision to trigger critical incident or not
  1: **procedure**
Decision Phase (I)
  2:    **label:** *Top*.
  3:    Input←DatafromAnalysis
  4:    Filtered_(*input*)_
← security-specific Incident
  5:    Classification_(*input*)_←trusttagsandseveritylevel
  6:    **if** Input **then**
  7:       Check Filtered${}_{\left(input\right)}$
  8:       Check Classification${}_{\left(input\right)}$
  9:       **if** Classify${}_{\left(input\right)}$ **then**
10:          Verify
11:          Report HMI Feedback
12:          Report to BlackBox
13:          Allow flow to Critical module
14:       **else**
15:          break
16:       **end if**
17:    **else**
18:       Return to *Top*
19:    **end if**
20: **end procedure**


### 5.2. Critical Incident Module-Based Controller

Similarly, in decision phase (II), data from the decision phase (I) also invoke a security check that involves (1) checking the validity of that specific message, (2) checking the classification attributes, associated tags, and markings related to trust and severity levels unique to individual components such as sensors, secure gateways, infotainment, and others, and (3) checking if each message has a threshold marker to sure the appropriate message is delivered to the required source-to-destination, and (4) checking the resolvability index. When each preliminary security step is passed, the classified messages are evaluated based on their threshold index against the acceptable pre-defined security index. If the threshold index exceeds the acceptable threshold, (1) the fallback mechanism is initiated, and (2) reporting is triggered to the HMI and black box. However, suppose the threshold index is lower than the threshold. In that case, the fail-over mechanism initiates instead of continuing the flow to decision phase (III) visible in Algorithm 2, Lines (7–16).
**Algorithm 2** Decision module for critical incidents.
   **Input**: Classified system incident with level
   **Output**: Critical level—decision to trigger fail-over/fallback mechanism
  1: **procedure**
Decision Phase (II)
  2:    **label:** *Top*.
  3:    Input_(*SI*)_
←ResultsfromSystemIncidence
  4:    Threshold_(*Critical*)_
←Threshold
  5:    Classification_(*input*)_
←trusttagsandseveritylevel
  6:    **if** Input${}_{\left(SI\right)}$ **then**
  7:       Check Classification${}_{\left(input\right)}$
  8:       Resolveability Index
  9:       **if** Classification${}_{\left(input\right)}$ >= Threshold${}_{\left(Critical\right)}$ **then**
10:          Trigger Fallback Mechanism
11:          Report HMI Feedback
12:          Report to BlackBox
13:       **else**
14:          Trigger Fail-Over Mechanism
15:          Report HMI Feedback
16:          Report to BlackBox
17:       **end if**
18:    **else**
19:       Return to *Top*
20:    **end if**
21: **end procedure**


### 5.3. Report/Support Module-Based Controller

Finally, in decision phase (III), invoked security measures include (1) checking the validity of specific messages for each input from the fail-over and fallback, (2) checking the classification attributes, associated tags, and markings related to trust and severity levels unique to individual components such as sensors, secure gateways, infotainment, and others, and (3) checking each resolved status messages from the fail-over and fallback.

Upon each preliminary security check that is passed, a resolved status indicating the true status from the fail-over indicates that the security incident with a threshold index lower than the acceptable threshold is resolved or compensated. On the other hand, a true status from the fallback also indicates that the initiated action has no errors or drawbacks, and the security operations are re-summable. In each of these procedures visible in Algorithm 3, Lines (11–14) and Lines (15–18), each critical security action and measure taken are reported to the black box and the HMIs for in-vehicle and remote management synchronizations. From Algorithm 3, Lines (19–22), a false resolvable status triggers or initiates the reporting/support process module, which is also visible in Figure 3. Archiving is also applicable at this stage of the process. In addition to the main functions of the decision modules illustrated, its functional scope is to help in-vehicle decision-making dynamically and proactively. This base contributes to the self-awareness criterion in its decision-level accountability of each critical process and action taken.
**Algorithm 3** Decision module for report/support.
   **Input**: classified critical-level decision
   **Output**: decision to trigger a report for support or not
  1: **procedure**
Decision Phase (III)
  2:    **label:** *Top*.
  3:    Input_(*fail-over*)_
← Results from Fail-Over mechanism
  4:    Input_(*fall-back*)_
←ResultsfromFallBackmechanism
  5:    Resolved_(*CI*)_
←IsIncidentResolved
  6:    Classification_(*input*)_
←Allocationtrustandseveritylevel
  7:    FBack/FOver←SameasFallBackandFailOver
  8:    **if** Input **then**
  9:       Check Classification${}_{\left(input\right)}$
10:       Check Resolved status
11:       **if** Input${}_{\left(fail-over\right)}$ and [Resolved${}_{\left(CI\right)}$ = True] **then**
12:          Report HMI Feedback
13:          Trigger BlackBox Report request
14:          Return to *Top*
15:       **else if** Input${}_{\left(fall-back\right)}$ and [Resolved${}_{\left(CI\right)}$ = True] **then**
16:          Report HMI Feedback
17:          Trigger BlackBox Report request
18:          Return to *Top*
19:       **else if** [FBack/FOver] and [Resolved${}_{\left(CI\right)}$ = False] **then**
20:          Trigger BlackBox Report request
21:          Trigger Remote Assistance
22:       **end if**
23:    **else**
24:       Return to *Top*
25:    **end if**
26: **end procedure**


## 6. Fail-Over

Failures and faults are critical aspects of autonomy through a predefined set of instructions because they can translate into the failure of the entire complex security system, which can even influence a complete system failure. The general understanding of a fail-over is protection from failure whereby a piece of standalone equipment automatically takes over in our security context. The procedure can entail an automatic transfer of control to a duplicate in faults or failure detection. However, implementing a fail-over for each security aspect of an autonomous vehicle may be practically non-applicable in most cases. Therefore, as shown in Figure 6, this specific security exploration considers three factors: control handoff, optimization management, and the utilization of decision modules in Figure 4.

The fail-over sub-modularization envisions an optimization management role responsible for compensating for security failures. This is to achieve and retain operational status if needed without entirely compromising the decision-making process and its reliant outputs. These outputs include decision-related modules, evaluation nodes, devices, and components, such as, for example, compensating for failures in a zonal sensor cluster due to deep neural adversarial intervention within a specific section or an area covered by two or more sensors for fusion-specific decision polling.

Furthermore, parameter checking and monitoring and their correspondent security trustworthiness are vital. Security-specific outputs via multiple interface configurations and calibration consistency are also necessary during decision-making with its trustworthy ratio. This fail-over security module is actively or passively ready to react to any failure with optimizations or fail-safe protocols. These procedures in the optimization management, for example, can include routing services, trusted external systems, identity, and access management security issues in either service, runtime parameters, or application programmable interfaces (APIs).

## 7. Fall-Back

The primary premise of a fall-back mechanism is a secondary system or procedure that is designed to activate in the event that the primary system fails or malfunctions; its existence in the security context for autonomous vehicles is critical. The vehicular state-of-the-art implements several fall-back mechanisms such as electronic stability control (ESC), brake override systems, transmission control modules (TCM), engine control modules (ECM), and others. However, security-specific measures attributed to cybersecurity measures rely on best practices, standards, and recommendations from governing bodies and vendor-specific measures. Therefore, exploring the security-specific fall-back mechanism in Figure 7 explores an applicable alternative procedure where the fall-back initiation or triggering depends on the decision module-based controllers.

This module has security role-based handoffs that address control, user, and operation center handoffs with a consensus module that considers identity, access, and decision logic. In contrast with the state-of-the-art, the vision of the decision logic along with the analysis module is archiving critical decisions with the black box, as in Figure 4. As referred to in Figure 4, the analytics process module handles the compute sections leading to decision evaluation. The passive/active system incident evaluation module, critical characterization module, and resolvability module with relation mapping for correspondent incident mitigation have assistive attestation through the event listers and message bus. The premise is security contingency; however, it also requires safeguards such as identity and access management control, which must match the consensus module upon in-vehicle or external initiations. In cases where failures prohibit such functionality through the critical-level decision module, a possible control handoff becomes an option if applicable in a security incident that is not internally resolvable or needs external intervention.

## 8. Visualization

The visualization interface module-based controllers in Figure 8, have two dimensions: the in-vehicle container dimension and the virtual security operation center(VSOC) container dimension, as well as corresponding management modules between the containers. It handles necessary visual and status indicators beneficial to occupants aided by the module-based agent controllers, which are communication carriers relaying data and information across and among other module-based controllers.

The VSOC container represents the future trends for intelligent, connected, and autonomous vehicles in the context of security as subscription-based models gradually become the norm. In SAE level 3 to level 5, autonomous vehicles by popular and leading companies such as Waymo, Tesla, and Baidu have features such as remote assistance, remote management, redundant controls, and safe pull-over. In their routine operations, some of these services require over-the-air updates, operational status, and other requirements and characteristics, making remote operational centers visualize the statistics in real time for security impacts. Human-to-machine interfaces are applicable for such situations with emphasis on our three assumptions.

The execution archive component in Figure 4 is the immutable security black box for keeping records of all necessary critical incidents that need to be kept based on the pre-defined rules within a time frame or range. Because of dynamic security communication risks resulting from the in-motion vehicle, factors such as safety, liability, privacy, and data security would be a concern that needs addressing. Both in-vehicle and external communications attributed to generated source and destination routine operations security trust-level risks for the sources’ identities and destinations have record keeping.

## 9. Discussion

In this manuscript, the self-aware cybersecurity defensive capability proposed aims at connected, intelligent, and autonomous vehicles in their vehicular security context. As a limitation, this solution does not claim to replace any automobile cybersecurity defense implementation or deployments by various vendors. However, we present an independent extension and interpretation of the state-of-the-art extrapolation from best practices, cybersecurity standards, and several practical solutions for connected, intelligent, and autonomous vehicles. Recognizing the importance of vehicular security concerns, we see cybersecurity dependency on vehicular autonomy prolificacy as a double-edged sword [55]. Its security intricacies and complexity benefit both the blue and red teams, white and black hats, enablers, and experts. It is because any security hallmark sophistication and susceptive piggy-backs with a temporal sliding window have probable dimensions in attack isolations and impacts. As cybersecurity measures such as secure boot process, intrusion detection and response, user authentication, and others already exist in some connected, intelligent, and autonomous vehicles, others such as isolated control systems, network segmentation, real-time threat detection and response, and unique and specific security system backups, such as some revealed in this manuscript, still needs research attention. Hence, a vehicular security motivational challenge in exploring, disassembling, and augmenting applications and custom technical deployments will vary between companies. Furthermore, carefully balanced, specially tailored, and specific automotive cybersecurity solutions are sometimes discernable rather than flooding the system with numerous solutions in the hope of capturing everything. It is also essential for possessive instantaneous deterministic probability in system analysis, self-resolvability, fail-over, fall-back, and critical reporting.

Misconfiguration is one of the most common security concerns and risks that often arises from complex heterogeneous systems with several layers with interlaced and unique functionalities. With Figure 4, Figure 5, Figure 6, Figure 7 and Figure 8, this security scheme can be implemented or deployed in several variations with the underlining concept and principle that misconfiguration can expose security implementations and infrastructures on multiple fronts.

Finally, the framework offers an iterative implementation approach for extensive testing and attestation parameters. This feature and its modularization by design also mean the granular disassembly of each component for further investigation needed in derivable solutions. Subsequent testing and experimentation can also be focused on the configurations and parameters while monitoring the output results for how it influences the investigation of security components concerning other components through execution traces, memory dumps, and syscall traces. With software and hardware awareness of system incidents and their severity through these security measures, the action taken in some routine operational activities for critical scenarios and cases, in addition to routine monitoring system activities and system logs in real-time, can help diagnostics in other realms beyond security.

## 10. Conclusions and Future Scope

In this work, we provided a systematic overview and exploration of security dynamism, self-awareness, and adaptiveness in the vehicular ecosystem. Based on our exploration, we proposed a hierarchical self-aware security architecture that effectively establishes accountability at the system level. In addition, we presented the sub-components of the self-aware architecture, their algorithms, and the integration of a security-specific black box. To this end, this work extended the existing state-of-the-art by a self-aware architecture that addresses security complexities and challenges concerning dynamism, adaptiveness, and self-awareness. Our future work will focus on the intricacies of implementing this architecture in an artificial intelligence environment.

## Figures and Tables

**Figure 1 sensors-23-08817-f001:**
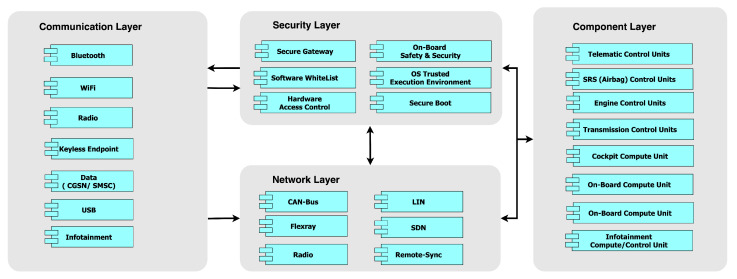
State-of-the-art vehicular security architecture.

**Figure 2 sensors-23-08817-f002:**
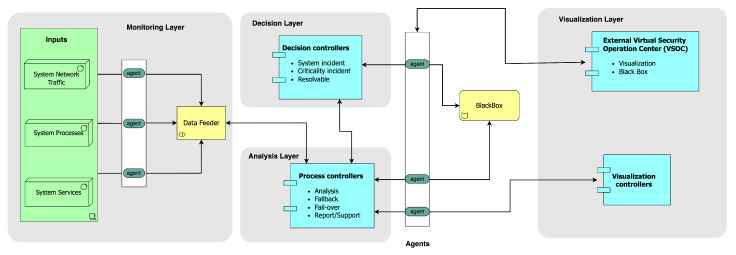
Overview of the proposed self-aware security architecture.

**Figure 3 sensors-23-08817-f003:**
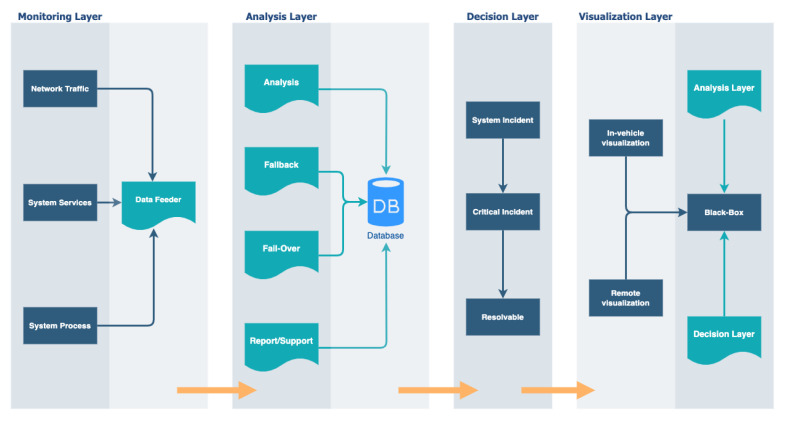
Self-aware architectural hierarchical phases.

**Figure 4 sensors-23-08817-f004:**
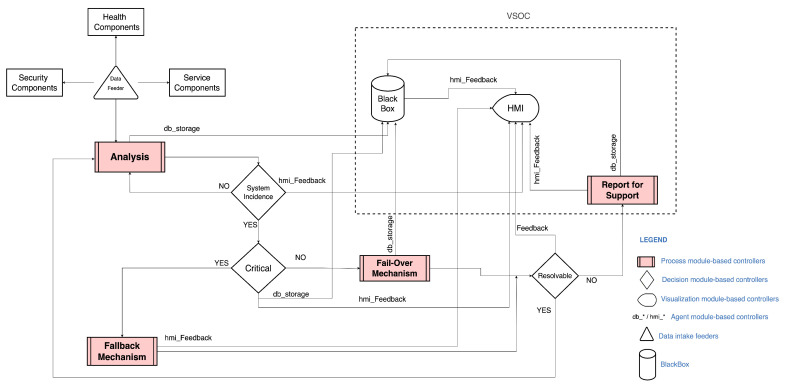
Internal transactions in the proposed self-aware security architecture.

**Figure 5 sensors-23-08817-f005:**
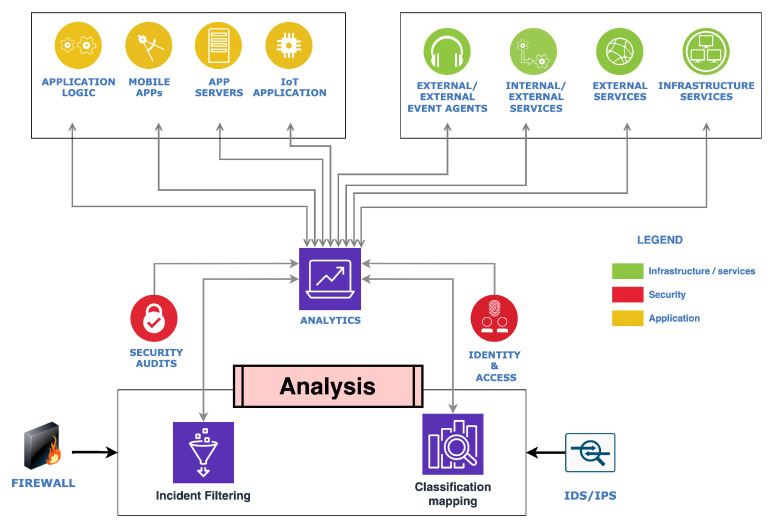
Analysis module-based controller.

**Figure 6 sensors-23-08817-f006:**
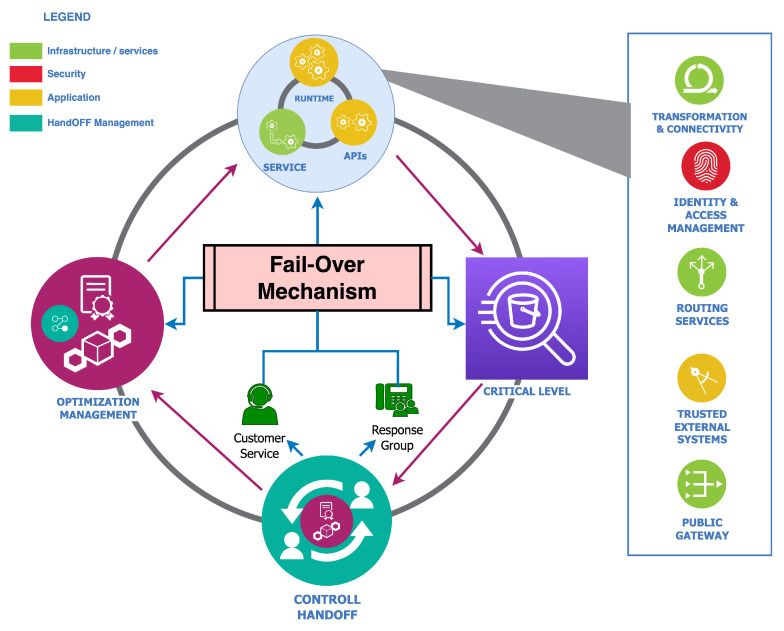
Fail-over module-based controller.

**Figure 7 sensors-23-08817-f007:**
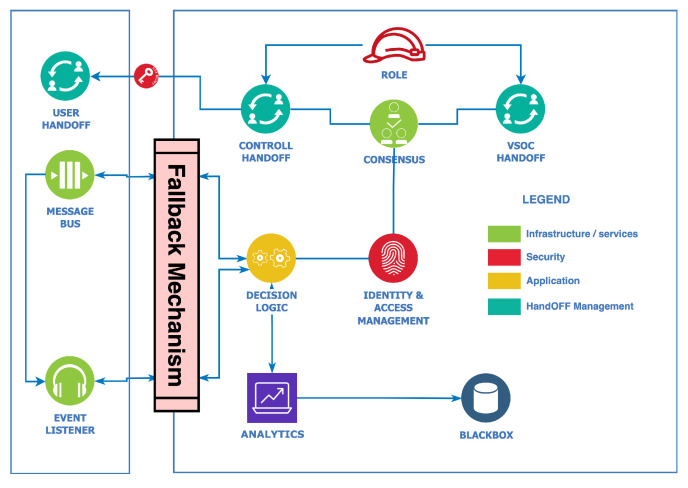
Fall-back module-based controller.

**Figure 8 sensors-23-08817-f008:**
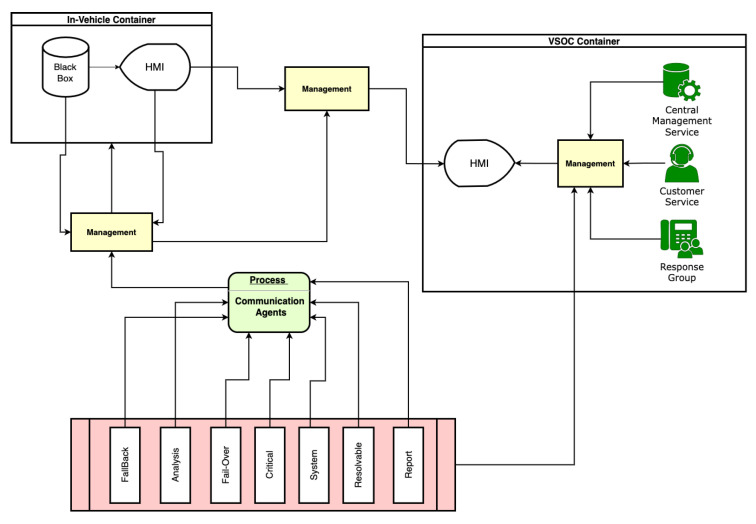
Visualization module-based controller.

**Table 1 sensors-23-08817-t001:** Examples of security measures in modern vehicles.

**Trusted execution environments (TEEs)** are utilized in low-energy embedded devices in addition to other cloud solutions and desktop computers as an isolated execution environment and platform [21]. Its implementation is to leverage secure application execution with the provided enclaves and integrity as a separate secure operating system (OS) in isolation specific to its hardware environment [22].
**Two-factor authentication (2F)** is a widely used provision of assurance of the claimed identity requiring additional attestations or proof. Two independent factors are involved through verification and authentication processes to enhance, restrict, or prevent unauthorized access [23] to services and client-side infrastructural resources such as vehicular remote applications and systems.
**White-listing** defines a set of rules or pre-selective applications, processes, or system activity permitted for execution at runtime or allowed to run when required [24]. Vendors deploy this solution in vehicles to restrict user space and system domain access. It also extends to hardware-based, native OS implementation and third-party add-ons [25].
**Secure boot** using a hardware–software combination ensures that all the software implementations are from a trusted source with vendor authentication to ensure that only the right OS is booted [26]. Secure boot, often combined with whitelisting, is used for restricting and applying selective installations of applications and firmware.
**Tamper-proofing** sensors are used in modern vehicles to detect various forms of tampering [27] in several sections, including the electronic control unit (ECU), engine, and other in-vehicle controls. It is a widespread security measure with external add-ons from third-party providers.
**Access control** is a common form of restricting unauthorized access to create roles according to the job functions performed to grant permissions (access authorization) to that role [28]
**Firewall** deployment prevents external attacks on the in-vehicle systems, such as the infotainment system. It is applicable in several ways in a vehicular context at the circuit level, application level, and filtering level [29].
**Intrusion detection systems** monitor in-vehicle networks for suspicious activities and are often combined with the other security measures mentioned in this table.

## Data Availability

Not applicable.

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
