# Peer review of "Self-Aware Cybersecurity Architecture for Autonomous Vehicles: Security through System-Level Accountability"

_sensors, 2023, doi:10.3390/s23218817_

Round 1

Reviewer 1 Report

Comments and Suggestions for Authors

Good introduction presenting a sound motivation for this proposal. In the background section an overview of current approaches to security in modern vehicles is given. As a preparation of the subsequent presentation of the newly proposed, extended architecture, a figure similar to the current figure 1 would be helpful. That figure showing the before-status of system design will then help the reader to understand the novelty and improvement of the proposed architecture.

The proposed architecture is overall well explained. The mentioned components also shown in figure 1 shall be marked in the figure to ease understanding. The logically imprinted layered approach of data processing is not shown in figure 1, however, this is already used in its explanation. The layers are only visualised in the much later referenced figure 2. There is room for improvement int the presentation of the structural and logical architecture.

The meaning of the arrows pointing back from the analysis layer to the components in figure 4 is not explained in the text. Figures 4 through 6 shall be placed in the subsection they belong to. The colour-coding of the components in figures 4 through 6 (why red, yellow and green?) are neither explained nor self-explanatory.

Overall, a sound architecture presentation. However, to me the improvement to existing architectures does not show clearly.

The proposed architecture is motivated, explained and discussed. The underlying work seems to be of purely theoretical nature; thus, the mayor shortcoming of the presentation is a missing evaluation. Consequently, the applicability of the proposed architecture remains to be evaluated. Nevertheless, sharing the proposed approach with the community is a good first step.

Comments on the Quality of English Language

There is a verb missing in line 89-91

There is a “section” missing in line 164.

In general, the readability will improve if the numerous numbered lists are actually using the typical layout for lists and are not just separated by the numbers in a plain paragraph.

Author Response

Good introduction presenting a sound motivation for this proposal. In the background section an overview of current approaches to security in modern vehicles is given. As a preparation of the subsequent presentation of the newly proposed, extended architecture, a figure similar to the current figure 1 would be helpful. That figure showing the before-status of system design will then help the reader to understand the novelty and improvement of the proposed architecture.

Ans: Thank you very much for these comments and contributions, we very much appreciate these reviews. Thank You. 

The proposed architecture is overall well explained. The mentioned components also shown in figure 1 shall be marked in the figure to ease understanding. The logically imprinted layered approach of data processing is not shown in figure 1, however, this is already used in its explanation. The layers are only visualised in the much later referenced figure 2. There is room for improvement int the presentation of the structural and logical architecture. 

Ans:  We have improved the figure 1. Now, the layers are labeled for easy visualization.

The meaning of the arrows pointing back from the analysis layer to the components in figure 4 is not explained in the text. Figures 4 through 6 shall be placed in the subsection they belong to. The colour-coding of the components in figures 4 through 6 (why red, yellow and green?) are neither explained nor self-explanatory. 

Ans:  All figures are placed in the subsection they belong to in the latex file. However, the rendering sometimes displaces them. Much effort is now being put into the content arrangement to reduce these effects. The colors in Figures 4 through 6 now have technical meaning with legends. They are now normalized with different colors to address this concern.

Overall, a sound architecture presentation. However, to me the improvement to existing architectures does not show clearly. 

Ans: The before-status of system design (baseline architecture) is now added as Figure 1 to show the improvement over existing architecture.

The proposed architecture is motivated, explained and discussed. The underlying work seems to be of purely theoretical nature; thus, the mayor shortcoming of the presentation is a missing evaluation. Consequently, the applicability of the proposed architecture remains to be evaluated. Nevertheless, sharing the proposed approach with the community is a good first step. 

Ans: The evaluation of the architecture is the next step, however, this architectural concept is tested in real life.

Reviewer 2 Report

Comments and Suggestions for Authors

Research article titled “Self-Aware Cybersecurity Architecture for Autonomous Vehicles: Security through System-Level Accountability” is good contribution and current demand of research trends. However, to be acceptable of it following suggestions should be incorporated in the revised manuscript:

Abstract and introduction should be limited by the limitations of the state-of-the-art techniques....motivation and contribution should be clearly stated...

Abstract must be modified accordingly; Start with Background, then add the scope and problem definition and then highlight what is the aim/objective of the article and what novelty is there in the proposed technique. In the last lines, highlight in what %age and in what parameters the proposed work is better as compared to existing techniques and what is the overall analysis of the technique.

In the literature, more recent references should be considered, as only a few references from 2021, 2022, and 2023. The same is needed to justify the novelty of the article. The following article may aid in the discussion section's identification of a significant research gap:

--Aabid Rashid, Sachin Kr Gupta, Zeba Khanam, Mamoon Rashid, Sultan S. Alshamrani, Ahmed Saeed AlGhamdi, “A Novel Approach for Securing Data against Adversary Attacks in UAV Embedded HetNet using Identity Based Authentication Scheme”, IET Intelligent Transport Systems, Wiley, 1-19, 2022.

More description is needed for figure 1 by considering technical aspect of it. Further few figures quality is very poor should be re-draw with high resolution.

In study which type of IDS/IPS system has been considered more details discussion is needed on which.

All algorithm must be presented in complete pseudorandom form by proper considering input and possible output of it.

Future extension of study should be added on conclusion section..

In results and discussion section, comparative analysis of results with existing techniques must be made..

Conclusion should be extended with Future Scope of the study…

Comments on the Quality of English Language

NA

Author Response

Research article titled “Self-Aware Cybersecurity Architecture for Autonomous Vehicles: Security through System-Level Accountability” is good contribution and current demand of research trends. However, to be acceptable of it following suggestions should be incorporated in the revised manuscript:

  1. Abstract and introduction should be limited by the limitations of the state-of-the-art techniques....motivation and contribution should be clearly stated…
  2. Abstract must be modified accordingly; Start with Background, then add the scope and problem definition and then highlight what is the aim/objective of the article and what novelty is there in the proposed technique. In the last lines, highlight in what %age and in what parameters the proposed work is better as compared to existing techniques and what is the overall analysis of the technique. 

Thank you very much for these comments and contributions. We appreciate this review very much. Thank You.        

Ans: Point 1 and 2 are now addressed and the Abstract, Introduction, and other aspects are updated based on the comments.

3. In the literature, more recent references should be considered, as only a few references from 2021, 2022, and 2023. The same is needed to justify the novelty of the article. The following article may aid in the discussion section's identification of a significant research gap:

4.    --Aabid Rashid, Sachin Kr Gupta, Zeba Khanam, Mamoon Rashid, Sultan S. Alshamrani, Ahmed Saeed AlGhamdi, “A Novel Approach for Securing Data against Adversary Attacks in UAV Embedded HetNet using Identity Based Authentication Scheme”, IET Intelligent Transport Systems, Wiley, 1-19, 2022.

Ans: Points 3 and 4 are now addressed and updated to reflect these comments:
1.    Recent references are now added from the 2021 – 2023 
2.    The structure of the abstract is amended to reflect starting with the background, scope, problems, aims/objectives, and novelty accordingly.   

More description is needed for Figure 1 by considering technical aspect of it. Further few figures quality is very poor should be re-draw with high resolution.

Ans: Figure 1 is the simplified version of figure 2 & 3, therefore all the technical details associated with Figure 2 & 3 is applicable to Figure 1. 

 In study which type of IDS/IPS system has been considered more details discussion is needed on which.

Ans: The article addresses the architectural design while the implementation of each of the layers will be addressed as our next step. Furthermore, there are no restrictions on IDS/IPS type with this specific architecture. Detailed discussion on IDS/IPS is covered in the implementation and evaluation which is beyond the scope of this article. 

 All algorithm must be presented in complete pseudorandom form by proper considering input and possible output of it.

Ans: Now, all the inputs and outputs of the algorithms are properly addressed.

Future extension of study should be added on conclusion section. In results and discussion section, comparative analysis of results with existing techniques must be made. Conclusion should be extended with Future Scope of the study.

Ans:  The above comments are addressed in the revised manuscript.

Reviewer 3 Report

Comments and Suggestions for Authors

The authors offer an overview of self-Aware Security Concept in connected, intelligent, and autonomous vehicles. They offer the hierarchical security architecture and the design of architectural components and algorithms with the integration of a Black-box. The topic is relevant.

1. Please add the literature review section.

2. In 4.4 the authors mention  three decision module-based controllers in the decision layer, but they don't describe them, please offer the details description.

3. The algorithm 2 and algorithm 3 mention classification stage, the authors don't explain clearly how the classification is done. Please explain the classification phase in details.

4. Please offer the more detailed description of Figure 7. "Visualization module-based controller",

5. Conclusion session is very poor, please extend the section and describe the conclusions properly.

6. Please explain why your method is better then the related approaches. Describe also the disadvantages.

Author Response

Thank you very much for these comments and contributions. We appreciate this review very much. Thank You.        

1. Please add the literature review section.

Ans: Literature review is added now to reflect comment

2. In 4.4 the authors mention three decision module-based controllers in the decision layer, but they don't describe them, please offer the details description.

3. The algorithm 2 and algorithm 3 mention classification stage, the authors don't explain clearly how the classification is done. Please explain the classification phase in details.

4. Please offer the more detailed description of Figure 7. "Visualization module-based controller”,

Ans: Comments on contributions 2, 3, and 4 are addressed according to the following:
1.    Each decision module has its subsection with a detailed description
2.    The classification is now extended to reflect how the classification is performed to address comment 3
3. algorithms 2 and 3 are reformatted to meet and address the first part of comment 3 
4.    Figure 7 is now extended further with more details to address comment 4

Conclusion session is very poor, please extend the section and describe the conclusions properly.

Ans: The conclusion section is updated to reflect the comment.

Please explain why your method is better then the related approaches. Describe also the disadvantages.

Ans: A literature review is added to show the added value of the proposed architecture compared to existing ones. The limitations of the method are also included in the research scope and conclusion.

Round 2

Reviewer 2 Report

Comments and Suggestions for Authors

now it may be accepted

Comments on the Quality of English Language

proof read required..

Reviewer 3 Report

Comments and Suggestions for Authors

The paper is improved